# Brain mass explains prey size selection better than beak, gizzard and body size in a benthivorous duck species

**Karsten Laursen** [1] *, **Anders Pape Møller**[2]

**1** Department of Bioscience, Aarhus University, Rønde, Denmark, **2** Ecologie Systématique Evolution, Université Paris-Sud, CNRS, AgroParisTech, Université Paris-Saclay, Orsay Cedex, France

* kl@bios.au.dk

**Data Availability Statement:** All relevant data are within the manuscript attached as a Supporting Information file.

## Abstract

Prey size selection in some bird species is determined by the size of the beak. However, we assumed for bird species swallowing whole prey that a cognitive process may be involved. As cognitive feature, brain mass was used. We hypothesized that the mass of the brain was more strongly positively correlated with prey size than morphological features such as beak volume, gizzard mass and body mass. We tested this hypothesis on eiders *Somateria mollissima* that swallow the prey whole, by using mean and maximum size of nine prey categories. Eiders were collected at the main wintering grounds in Denmark. As index of brain mass we used head volume, which is positively correlated with brain mass ($r^2 = 0.73$). Head volume of eiders was significantly, positive correlated with mean and maximum size of blue mussels *Mytilus edulis*, razor clams *Ensis directus* and all prey sizes combined and the maximum size of draft whelk *Hinia reticulata* and conch *Buccinum undatum*. Gizzard mass was also significantly positively correlated with maximum size of draft whelk and conch. Beak volume and body mass was not significantly correlated with the size of any of the nine food items. Analyses of effect size for organs showed that head volume was positively related to prey size, whereas beak volume, gizzard mass and body mass did not show a significant positive relationship. These results indicate that cognitive processes connected to brain mass may be involved in prey size selection by eiders.

## Introduction

Beak morphology of some bird groups has evolved through selection and been shaped to utilize and ingest certain types of food as already described by Darwin [1] and confirmed in later studies [2,3]. Taking finches as an example, most beaks of the species are formed to utilize food resources within certain habitats and to crush seeds of different hardness [2–4]. Within species, individual beak may also determines prey size selection [5,6]. However, large scale comparative studies showed that diet only explains a small fraction of variation in the shape of beaks among species and the morphology of beaks may be controlled by non-dietary factors [7,8]. A recent study of a shorebird, the red knot *Calidris tenuirostris*, revealed that prey size

**Funding:** This work was supported by The 15. June Foundation, Denmark (Ref. 2015-B-132 and 2019-J-3) received by KL (http://www. 15junifonden.dk/). The funder had no role in any aspects of the study or preparation of the manuscript.

**Competing interests:** The authors declared that no competing interests exists.

was correlated with the size of the gizzard, and not with the size of the beak [9]. When eating, knots swallow the entire prey e.g. bivalves and snails, which are crushed in the gizzard. Obviously, the gizzard is unable to make any assessment of the size of a prey item before it ends in the gizzard, which implies that a decision mechanism may be involved before the prey is taken. The decision mechanism may involve sensory, neural and cognitive structures and active interactions between these [10,11]. In the knot, the mechanism may involve feedback from the gizzard given information, e.g. that a given prey is small, another prey item is suitable in size, and a third is too big. Thus, through a series of foraging bouts, a bird individual may adjust its foraging to the most optimal size in relation to gizzard size, given the condition of the present food supply and prey species composition in the environment [10]. Otherwise, the size distribution of prey in the gizzards should be randomly representing the size distribution of benthos in the sea bottom rather than the size of the gizzard. Comparisons of the prey size taken and the size of benthos on the sea floor show that size distribution of the prey taken is not random, indicating that a selection process is involved [12–14]. Large prey items are selected to optimize food intake and improve body condition [9,15]. We suggest that the brain must be a part of this mechanism. Although the suitability of brain mass as a measure of cognitive abilities is debated, there seems to be an acceptance that brain mass clearly influences self-regulation and performance within taxonomic groups [16]. Brain mass and its cognitive functions is associated with morphological evolution in birds, and is generally positively correlated with behavioral patterns including foraging, learning abilities, behavior in complex social environments and flexibility in exploiting new habitats [11,17–22]. Thus, survival and flexibility of species are suggested to be connected to brain mass or the functions associated with it [21–24]. Further, birds with large brains take appropriate decisions when a potential threat is approaching, and they are able to avoid dangerous situations in contrast to species with small brains [25,26].

We hypothesize that individuals with large brains are taking larger prey compared to individuals with smaller brains. Thus, we expect that individuals with large brain mass take appropriate decisions during foraging and select prey of optimal size in relation to gizzard mass. From this follows that the correlation coefficient for brain mass in relation to prey size is larger than for non-neural anatomical structures such as beak volume, gizzard mass and body mass. These anatomical structures together with sex were included in the analyses as for some bird species these traits are known to have an effect on prey size selection [4,24]. As brain mass measure, we used head volume [27]. Head volume, together with gizzard mass, beak volume, body mass and sex were analyzed in relation to mean and maximum size of prey species and groups of prey in the gizzard. We tested the hypothesis on another benthos eating bird species, the eider *Somateria mollissima*, during the non-breeding season. The eider like the knot, swallows the entire prey, which is crushed in the gizzard. Furthermore, in both the knot and the eider gizzard mass changes in relation to season and feeding conditions, which may complicate selection of prey of a given size [15,28–30]. Studies of eiders at the wintering grounds in Denmark show that large gizzards contain large prey items, and that large gizzards are related to superior body condition and successful reproduction [15,29]. These results imply a relationship between selection of large prey, gizzard mass, body condition and finally successful reproduction. The cognitive competences during foraging may involve social abilities and risk assessment as demonstrated for eiders staying in the Wadden Sea despite of hunting activity feeding on blue mussels, their preferred food item [31]. To mitigate being located by hunters and shot, eiders reduced flock size according to hunting intensity. Eiders are able to locate and concentrate in marine sites with high food production, high mussel quality and mussel stocks or high benthos biomass [29,32–34]. However, studies of cognitive abilities in relation to prey size are still missing.

Gape width is important for bird species that swallow food items whole, as for fruit-eating species [35,36]. This may also be the case for eiders. Thus, we examine an alternative hypothesis, that gape width restricted the maximum size of food items taken by eiders.

## Materials and methods

A total of 198 eiders, named the main sample, were collected between 10th February and 10th March, 2016–2019, by institutional staff from Aarhus University in Kattegat, situated in the central part of Danish waters (55˚ 50' N; 10˚ 20' E), under licenses from the Ministry of the Environment. The sample consisted of 108 males (104 adults, 4 juvenile) and 90 females (71 adult, 19 juvenile). The number of eiders collected in the four years were 42, 69, 45, and 42. When shot, the eiders were labeled with date and locality. They were frozen the day or the morning after being collected. A smaller sample, named the supplementary sample, was taken in February 2021 consisted of 21 eiders, 17 males and 4 females.

### Morphological variables and prey size

In the laboratory, for the main sample, we recorded body mass with a balance to the nearest 100 g. Information on sex and age (sexually mature or juvenile) were recorded using standard criteria [37]. Gizzard mass without content was measured on a balance to the nearest 0.1 mg. The length, height and width of the head were measured with calipers to the nearest 0.1 mm. Head volume (y) was estimated from the equation for an ellipsoid (y = (4/3 x $\pi$ x (head length–beak length/2) x (head height/2) x (head width/2)) [25]. The length, height, width of the beak were measured with calipers to the nearest 0.1 mm. Beak volume (y) was estimated from the equation (y = (4/3 x $\pi$ x (beak length/2) x (beak height/2) x (beak width/2)). The skull of 15 eiders was opened and the brain desiccated and weighed on a balance to the nearest 0.1 mg. The esophagus and gizzards were opened and the content was separated into nine categories (*Mytilus edulis*, *Cerastoderma edule*, *Ensis directus*, bivalves spp., *Littorina littorea*, *Hinia reticulata*, *Buccinum undatum*, *Carcinus maenas*, and other species) using the methods described elsewhere [29]. Due to a large number of broken items we could not be sure that all fragments belonged to the species named, parts of the items may belong to related species. The size of bivalves and snails were measured as the total (longest) length of the shell and for crabs the width of the carapace. For fragments of food items, the following criterion was used for quantification: For mussel, umbos were used supplied with the distal parts of the shells; for gastropods the columns; and for crabs the number of claws of the same size were counted and divided by four to get the number of crab individuals. Size of intact and broken food items was approximated to the nearest 5 mm. For the broken items we used a size-appropriate reference from a collection of intact prey samples. For each eider, the mean length and the maximum length of all categories of food items were estimated. For the supplementary sample, gape width and gape height was measured using calipers to the nearest 0.1 mm.

### Statistical analyses

For the main sample, the relationship between the mean and maximum size of the prey species and prey groups together with the mean and maximum size of all prey combined (as dependent variables) were analyzed in relation to beak volume, gizzard mass, head volume, body mass, sex and year (as explanatory variables) by use of a multivariate Generalized Linear Model (GLM). Year was included in the model as a categorical variable to account for variation and different sample size among years. Effects of beak volume, gizzard mass, head volume, body mass and sex (as explanatory variables) on the size of prey species were examined in a multivariate GLM analysis with signed effect size estimated as Pearson's correlation coefficient

as dependent variable. All variables stayed in the model for the test. In the multivariate GLM analyzes we assumed a normal distribution with an identity link function. JMP version 10.0 was used for the statistical analyze [38]. To reduce variance in all variables $\log_{10}$-transformations were used.

## Results

Of 1299 prey items in the main sample, blue mussels accounted for 28.3%, cockle 3.2%, razor clam 2.2%, other mussels 6.6%, periwinkle 10.4%, draft whelk 31.1%, conch 5.1%, shore crab 10.1% and other items 3.0%. The mean (and maximum size) of prey in gizzards varied from 9.5 mm (32 mm) in periwinkle to 102.7 mm (148 mm) in razor clam (Table 1 and Fig 1).

A positive relationship was found between head volume and brain mass ($F = 39.21$, $df = 1,13$, $p < 0.0001$, $r^2 = 0.73$). Head volume was positively correlated with mean and maximum size of blue mussel, razor clam and all prey combined and the maximum size of draft whelk and conch (Table 2 and Fig 2). Gizzard mass was significantly positively correlated with maximum size of draft whelk and conch and negatively correlated with mean size of cockle. Females took significantly smaller mean and maximum size of razor clams and conch than males. Beak volume and body mass was not significantly correlated with the size of any of the nine prey items.

Analyses of effect sizes for organs showed that head volume had a significant, positive effect on prey size ($\chi^2 = 23,7530$, $df = 1,65$, $p < 0.0001$, estimate (se) = 0.1026 (0.0193)), whereas beak volume, gizzard mass and body mass did not have significant effects.

The mean (se) gape width was 44.4 (0.6) mm and gape height 39.3 (0.9) mm. For five benthos species with the longest body length (see Table 1), relationships between body length and body width were measured, using the reference collection of intact benthos species. The length of the largest specimen of blue mussels in the diet of the main sample was 66 mm, with a corresponding width of 28 mm, for cockles 45 mm with a corresponding width of 42 mm, for razor clam 148 mm with a corresponding width of 22 mm, for conch 68 mm with a corresponding width of 40 mm and for shore crab 55 mm with a corresponding width of 38 mm. Comparisons between gape width and the width of the largest individual of the five benthos species found in the diet of eiders showed that they all were smaller than the width of the gape (44.4 mm compared to 28, 42, 22, 40 and 38 mm). Thus, the dimension of the gape seems not to restrict the size of the food items taken by eiders in this study. The alternative hypotheses are not considered further.

**Table 1. Statistics for prey size (mean (SE) and maximum size, mm) of food items in gizzards of eiders separated into eight categories: Blue mussel *Mytilus edulis*, cockle *Cerastoderma edule*, razor clam *Ensis directus*, bivalve spp., periwinkle *Littorina littorea*, draft whelk *Hinia reticulata*, conch *Buccinum undatum* and shore crab *Carcinus maenas*.**

| Prey species/group | N | Mean (SE), mm | Maximum, mm |
|---|---|---|---|
| Blue mussel | 123 | 27.4 (1.5) | 66 |
| Cockle | 25 | 27.4 (1.6) | 45 |
| Razor clam | 22 | 102.7 (4.0) | 148 |
| Bivalve spp. | 38 | 34.7 (1.6) | 58 |
| Periwinkle | 32 | 9.5 (1.2) | 32 |
| Draft whelk | 21 | 11.7 (1.2) | 28 |
| Conch | 12 | 42.2 (3.7) | 68 |
| Shore crab | 50 | 33.2 (1.0) | 55 |

N = number of eiders.

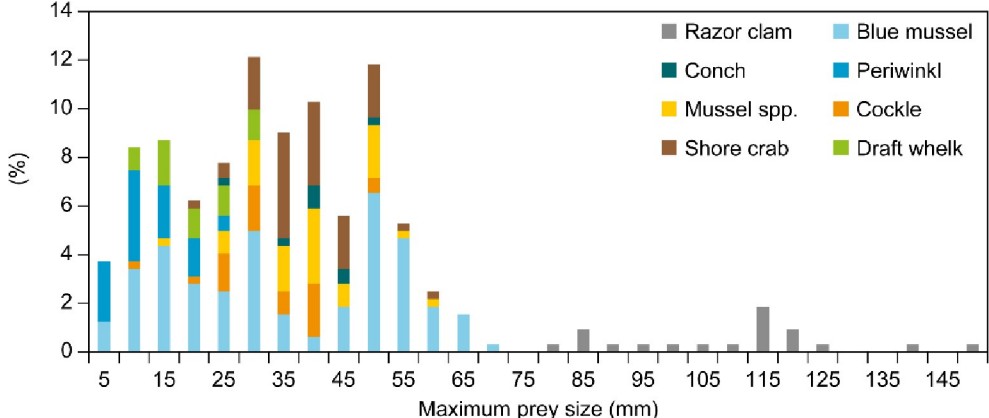

**Fig 1. Percentage (%) of maximum prey size (mm) of eight prey groups in the gizzard of eiders during winter in Danish waters.**

## Discussion

Individual eiders with large heads caught larger blue mussel, razor clam, draft whelk, conch and all prey items combined, than individuals with smaller heads. Head volume showed a large positive effect of prey size selection in eiders compared to beak volume, gizzard mass and body mass. These findings for birds support that foraging involve cognitive competences, as argued by Stephens et al. [10]. Beak size, estimated as beak volume, has in some bird groups been considered an important predictor for prey size selection, but a significant correlation with beak volume was not found for any prey items. These results support previous findings

**Table 2. Results of multivariate GLM analysis with size (mean and maximum size) of nine prey species or prey species groups as dependent variable and beak volume, head volume, gizzards mass, body mass, sex of eiders together with year as explanatory variables.**

| | | Beak volume | | Head volume | | Gizzard mass | | Body mass | | Sex | | Year |
|---|---|---|---|---|---|---|---|---|---|---|---|---|
| | N | p | Est. | P | Est. | P | Est. | p | Est. | p | Est. | p |
| Blue mussel, mean | 123 | 0.246 | -0.393 | **0.010** | 1.080 | 0.090 | 0.569 | 0.461 | 0.719 | 0.079 | -0.058 | 0.191 |
| Blue mussel, max. | 123 | 0.366 | -0.331 | **0.017** | 1.073 | 0.265 | 1.073 | 0.550 | 0.628 | 0.122 | -0.056 | 0.338 |
| Cockle, mean | 25 | 0.967 | -0.012 | 0.075 | 0.843 | **0.026** | -0.864 | 0.550 | -0.667 | 0.068 | -0.053 | **0.033** |
| Cockle, max. | 25 | 0.410 | 0.263 | 0.071 | 0.934 | 0.228 | -0.491 | 0.132 | -1.866 | 0.058 | -0.060 | 0.088 |
| Razor clam, mean | 22 | 0.582 | -0.174 | **0.014** | 0.451 | 0.170 | -0.176 | 0.429 | -0.428 | **0.014** | -0.050 | 0.393 |
| Razor clam, max. | 22 | 0.719 | -0.126 | **0.020** | 0.473 | 0.230 | -0.148 | 0.500 | -0.405 | **0.030** | -0.048 | 0.577 |
| Bivalve spp., mean | 38 | 0.812 | 0.120 | 0.244 | 0.404 | 0.284 | 0.345 | 0.203 | -1.104 | 0.913 | -0.003 | 0.719 |
| Bivalve spp. max. | 38 | 0.624 | -0.219 | 0.150 | 0.445 | 0.511 | 0.186 | 0.232 | -0.367 | 0.308 | -0.267 | 0.662 |
| Periwinkel, mean | 32 | 0.592 | 0.345 | 0.879 | 0.127 | 0.348 | 0.613 | 0.736 | -0.910 | 0.756 | 0.019 | 0.219 |
| Periwinkel, max. | 32 | 0.719 | 0.224 | 0.601 | 0.421 | 0.191 | 0.832 | 0.614 | -1.314 | 0.903 | -0.007 | 0.232 |
| Draft whelk, mean | 21 | 0.970 | 0.016 | 0.084 | 1.110 | 0.148 | 0.831 | 0.805 | 0.298 | 0.646 | 0.019 | 0.086 |
| Draft whelk, max. | 21 | 0.940 | 0.030 | **0.019** | 1.431 | **0.022** | 1.262 | 0.644 | -0.516 | 0.888 | 0.005 | **0.003** |
| Conch, mean | 12 | 0.534 | 0.371 | 0.055 | 0.992 | 0.261 | 0.646 | 0.640 | 0.652 | **0.042** | -0.077 | 0.599 |
| Conch, max. | 12 | 0.804 | -0.139 | **0.050** | 0.966 | **0.035** | 1.227 | 0.223 | 1.650 | **0.014** | -0.092 | 0.730 |
| Shore crab, mean | 50 | 0.401 | 0.152 | 0.445 | -0.135 | 0.064 | 0.238 | 0.862 | -0.064 | 0.788 | -0.004 | **< 0.001** |
| Shore crab, max. | 50 | 0.179 | 0.248 | 0.350 | -0.163 | 0.193 | 0.166 | 0.224 | 0.438 | 0.190 | 0.018 | **0.002** |
| All prey, mean | 198 | 0. 692 | 0.454 | **0.044** | 2.792 | 0.325 | 1.027 | 0.487 | -2.155 | 0.150 | -0.041 | **< 0.001** |
| All prey, max. | 198 | 0.740 | 0.393 | **0.040** | 2.948 | 0.283 | 1.156 | 0.444 | -2.452 | 0.648 | -0.050 | **< 0.001** |

*df* = 1 for beak volume, head volume, gizzard mass, body mass and sex; *df* = 3 for year. Significant variables are shown in bold font.

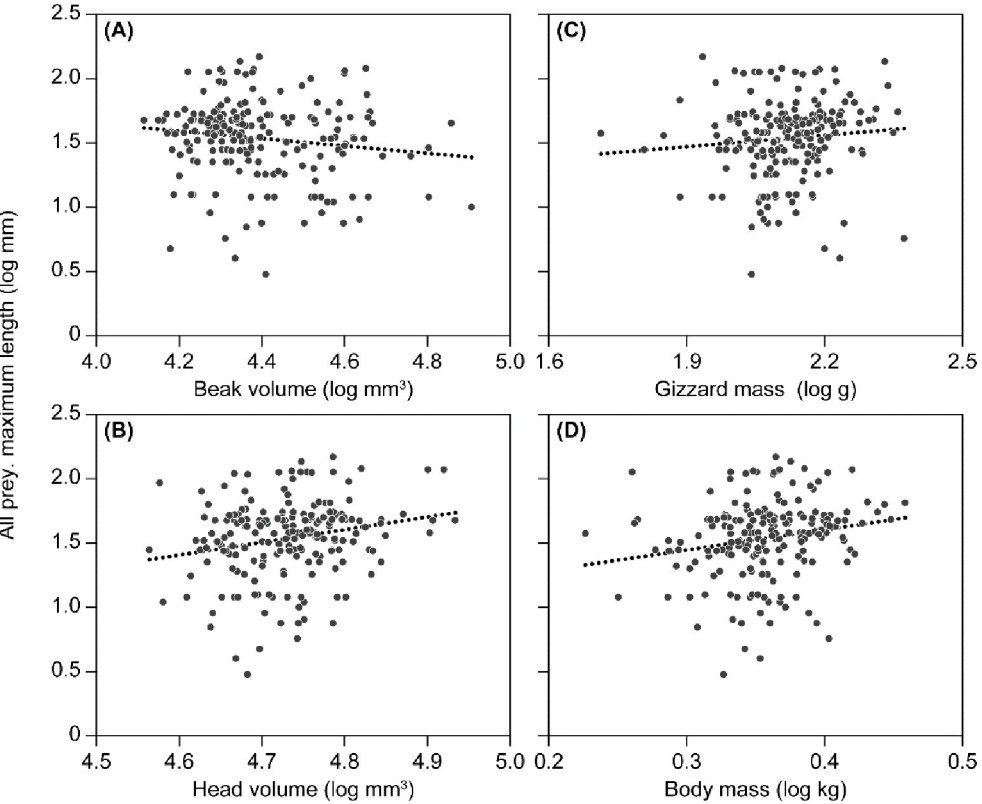

**Fig 2.** Relationships between log-transformed maximum prey size (mm) in gizzards of eiders and (A) log-transformed beak volume (mm³), (B) log-transformed head volume (mm³), (C) log-transformed gizzard mass (g) and (D) log-transformed body mass (kg). Regression lines are only for illustrative purposes.

that, in general, other morphological traits than beak size drives prey size selection [7,8,35,36]. Differences in prey size of males and females were found only for razor clam and conch, the two largest prey species (Table 1). Sex differences in prey size taken is often considered a mechanism to reduce competition between males and females [39,40]. The results indicate that sex related resource competition is modest in eiders. The reason for females taking relatively large prey could be that they are under strong pressure for building up body condition in winter due to the subsequent breeding season [29,39].

The study was undertaken in winter, when eiders are building up body stores for spring migration and the subsequent breeding season, implying that they were under time constraints and supposedly demonstrate particularly efficient feeding [41]. During this period, they accumulated about half of the body stores needed for the subsequent breeding. The other half is taken up, off the breeding sites before females enter the breeding grounds [42,43]. During the non-breeding season, the eider is a marine species that stays off-shore, foraging in flocks of up to 10,000 individuals [44]. Eiders are generally feeding at water depths of 4–8 m, and occasionally down to 20 m or even deeper [44]. Diving ducks use feet and wings to dive to the seabed where food is located [45]. Eiders with large webbed feet are obviously more efficient at diving and staying at the bottom while feeding, since individuals with large webbed feet have better body condition, than eiders with small webbed feet [46]. At the seabed, blue mussels are attached at mussel banks and visible, while cockles, razor clams and other bivalves are burrowed into the sea-floor. Most of the eider prey items are sessile, but able to move slowly, only shore crabs being the exception [39]. At the seabed, the eider locates the prey species and choose

which prey item and prey size to take. The bivalves burrowed in the sediment can be exposed by use of the feet [12]. Both shell sizes of the visible blue mussel and the burrowed razor clam were positively correlated with head volume. These two bivalve species are among the most important food items, when eiders build up body stores at the wintering grounds [29].

Eiders took large prey sizes, for species such as blue mussels up to 66 mm and for razor clams up to 148 mm. Razor clams of that size was larger than the length of the gizzard and while one part of razor clam was in the gizzard starting to be crushed and dissolved, the other part was in the esophagus. Small food items as periwinkle did not show any significant correlation with head size or any other organs. Comparable results were found in sympatric finch species and in fruit eating bird species showing no correlation between small size food items and anatomical structures as the size of beak, body or gape [3,35]. Large prey, especially large blue mussels, are associated with large gizzard mass, superior body condition and high reproductive potential in eiders [15]. However, experimental studies of eiders show that they select blue mussels with shell length of 10–20 mm due to high flesh to shell ratio in small mussels [47,48]. In Danish waters under natural conditions, blue mussels between 30–40 mm were preferred and individuals up to 80 mm were found in gizzards of eiders [49]. The size preference of blue mussels of 30–40 mm in size is confirmed in the German Wadden Sea, and it was argued that this size class was energetically the most profitable [12]. It is possible that fractions of small mussels and snails in our study were overlooked due to the large amount of shells from bigger individuals. In addition, the smaller prey may be crushed and dissolved more quickly than larger prey. This could lead to an overestimation of mean size of prey in our study. One the other hand, the size interval for blue mussels reported [12,49] are within the range of the mean size and the maximum size found in our study under natural conditions. These studies examined eiders under natural conditions, which could influence the prey size taken compared to experimental conditions.

Seasonal variation in resources and conditions imply that eiders seem to be able to take internal and environmental conditions into account when making decisions in relation to foraging and building up body condition for breeding, which are supposed to be related to cognitive abilities. These types of decisions are fundamental drivers of population dynamics [10].

## Conclusions

The results show that foraging eiders choose food items based on head size rather than morphological traits such as gizzard mass, body mass or beak volume. Since head size is positively correlated with brain mass, these results suggest that cognitive abilities expressed by brain mass are involved when eiders select prey sizes.

## Supporting information

**S1 File.**
(XLSX)

## Acknowledgments

We thank Johannes Erritzøe for desiccating the heads of eiders and two anonymous referees for constructive comments.

## Author Contributions

**Conceptualization:** Karsten Laursen.

**Data curation:** Karsten Laursen, Anders Pape Møller.

**Formal analysis:** Anders Pape Møller.

**Funding acquisition:** Karsten Laursen.

**Project administration:** Karsten Laursen.

**Writing – original draft:** Karsten Laursen.

**Writing – review & editing:** Karsten Laursen, Anders Pape Møller.

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
