## [Decision Letter · Decision Letter 0]

3 Nov 2020

PONE-D-20-17519

Brain size explains prey size selection better than beak, gizzard and body size in a benthivorous duck species

PLOS ONE

Dear Dr. Laursen,

Thank you for submitting your manuscript to PLOS ONE. After careful consideration, we feel that it has merit but does not fully meet PLOS ONE’s publication criteria as it currently stands. Therefore, we invite you to submit a revised version of the manuscript that addresses the points raised during the review process.

We look forward to receiving your revised manuscript.

Kind regards,

Vitor Hugo Rodrigues Paiva

Academic Editor

PLOS ONE

Additional Editor Comments:

Please carefully follow recommendations of both reviewers, specially those raised by reviewer 1 related with better displaying/ re-writting the study hypothesis and contextualisation of the work background during the introduction.

Journal Requirements:

2. In your Methods section, please provide additional location information of the collection sites, including geographic coordinates for the data set if available.

Reviewers' comments:

Reviewer's Responses to Questions

**Comments to the Author**

1. Is the manuscript technically sound, and do the data support the conclusions?

Reviewer #1: Yes

Reviewer #2: Partly

2. Has the statistical analysis been performed appropriately and rigorously? 

Reviewer #1: Yes

Reviewer #2: I Don't Know

3. Have the authors made all data underlying the findings in their manuscript fully available?

Reviewer #1: Yes

Reviewer #2: Yes

4. Is the manuscript presented in an intelligible fashion and written in standard English?

Reviewer #1: Yes

Reviewer #2: Yes

5. Review Comments to the Author

Reviewer #1: Comments for authors

Major comments

1. Introduction could be more streamlined and trimmed

Firstly, authors should try to be more clear in the text about separating which traits may be key for driving prey size selection at inter- and intraspecific levels. This could be pertinent for their own results as well. Also, I think, that Introduction’s first paragraph could be much more condense. Especially, I doubt if so much information about beak-related prey selection is really needed. In any case, it would be also great if you could be more specific and mention in the introduction about how well (less controversial as of recently) brain size is related to cognitive abilities. Lastly, information on study species would feel more natural when coming after presenting main hypothesis (lines 99-101).

2. Hypotheses

Authors may disagree, but as the hypothesis is written (99-101) it is not very well suited for the given study and more suitable for between seasonal comparison. Also, predictions are not well linked to hypothesis, but rather the results.

3. Statistical analyses

Could you please clarify if in your analyses a relative rather than absolute brain size was used? Brain size is often corrected for body size as larger individuals tend to have larger heads. You seem to have used body mass in the same model, but it is not fully clear to me if body size was left in the model as covariate. Based on the result table it probably was and no model selection was done, but could you clarify that. Would you get a different outcome by using structural size estimate rather than body mass for scaling?

Would there be a reason to suspect that YEAR needs to be included in the model (sampling areas, food availability etc.)? If not, would be nice to see explanation included in the statistical analysis description.

Would using median for prey size rather than mean influence the results? Median value would perhaps be more appropriate in relation to „optimal“ selection as it would indicate that individual selected such item more frequently.

As authors already have presence-absence data it could be interesting to see how prey diversity is related to brain size. Are bigger brained individuals going the diversification or the specialization route?

4. Discussion

I failed to find any comments on possible sex-related differences in prey size selection. Also, what about beak volume, especially given the attention to this trait in the introduction. Please, place your results in relation to your hypothesis and introduction a bit more clearly than it is now.

Minor comments:

Line 31: example of Calidris in the abstract is confusing. This specificity perhaps here is not needed.

Lines 40-43 This is confusing sentence as it is not clear what were dependent variables in your study. As it is now written it seems that you looked at brain size vs other morphological traits.

Line 46: The last sentence of the abstract feels disconnected from previous description of results.

Line 80 sentence is repetitive

Line 83 Something is missing here. Perhaps a connector part on how cognitive abilities are increasingly recognized to be associated with brain size?

Line 95-98 a confusing sentence

Line 108 „..on the other“

Line 115 what is the trait „potential“? Should it be „potential for successful reproduction..“?

Line 119 strange confusing sentence that looks to be clipped from Methods section. Few lines down the same information is mentioned.

Line 126 Please mention also if the two sexes (plus juveniles) were roughly equally represented for different years.

Line 132 what do you mean for age here (juveniles vs sexually mature)? Or was the chronological age recorded?

Line 136 Please, double-check if the provided reference (10:Zhang et al.) is correct for the formula. Zhang et al. did not measure head volume in their study. Also, check the formatting for the head volume formula.

Line 139 More appropriate term for „head“ here would be „skull“. I agree with authors that it could make sense to measure dry brain weight. However, also in previous studies Jaatinen et al. 2019 wet (?) brain mass was measured with the same results. Perhaps mentioning why dry brain weight was more appropriate option (if that was so), how was it standardized and that there were infact no qualitative differences may be good (in case it is so).

Line 168 What statistical programme was used for data analyses?

Line 202 check typo „ether“

Lines 205-207 These are repetition from result section. Could be better to introduce a general pattern or main finding.

Lines 216-222 This information actually would have been nice already in the introduction and not here.

Line 236 Apologies, but again a confusing sentence. Do you have in mind that you found extreme sizes or that eiders seemed to prefer larger than expected prey?

Line 238 what species was this done on? Also word „such as “ missing

Line 254 but you do not find that. If that is because lab vs natural condition then connection to the previous sentence should be clearer.

Line 255-258 very unclear sentence

Line 259-267 I would try placing this para just before conclusions.

Line 275 and 285 Would suggest to avoid „obviously“ and tone down.

Reference list not formatted properly.

Reviewer #2: The authors should change the title of the paper. For instance, their findings are not only positive relationship between brain volume and prey size but also positive relationship between the size of shore crabs and the area of web of feet.

The authors found positive relationship between head volume and brain mass (F = 39.21, df = 1,13, p < 0.0001, r2 = 0.73). But, can the head volume really be a proxy of the brain size? I agree that the head volume could be one of the indicators of the degree of brain development, though. Better idea would be that the authors use the volume of brain itself (or the size of brain domain specific for visual cognition) as the indicator of cognitive abilities.

Introduction Line3-7: Accipritridae, Podicipedidae, and Fringillidae, for example, are all "Family" name of birds. So, the authors should write them in regular style, not in italic style.

6. PLOS authors have the option to publish the peer review history of their article (what does this mean?). If published, this will include your full peer review and any attached files.

Reviewer #1: No

Reviewer #2: No

---

## [Author Response · Author response to Decision Letter 0]

17 Dec 2020

Reviewer #1: Comments for authors

Major comments

1. Introduction could be more streamlined and trimmed

Firstly, authors should try to be more clear in the text about separating which traits may be key for driving prey size selection at inter- and intraspecific levels. This could be pertinent for their own results as well. Also, I think, that Introduction’s first paragraph could be much more condense. Especially, I doubt if so much information about beak-related prey selection is really needed. In any case, it would be also great if you could be more specific and mention in the introduction about how well (less controversial as of recently) brain size is related to cognitive abilities. Lastly, information on study species would feel more natural when coming after presenting main hypothesis (lines 99-101).

-We now distinguish clearly between interspecific and intraspecific relations, see lines 56-61. In addition we have shortened the introduction, by deleting some of the text on beak-related selection, see lines 56-65 (in the original manuscript) and lines 55-61 in the revised version. 

-Further, we have added some sentences about the debate of using brain size as a measurement of cognitive abilities, see lines 80-82. 

-Lastly, there may be a misunderstanding, because information on the study species, the eider, is presented after (see lines 101-103) the hypothesis (see lines 91-92). 

2. Hypotheses

Authors may disagree, but as the hypothesis is written (99-101) it is not very well suited for the given study and more suitable for between seasonal comparison. Also, predictions are not well linked to hypothesis, but rather the results.

-We agree, and have changed the hypothesis, see lines 91-92. 

3. Statistical analyses

Could you please clarify if in your analyses a relative rather than absolute brain size was used? Brain size is often corrected for body size as larger individuals tend to have larger heads. You seem to have used body mass in the same model, but it is not fully clear to me if body size was left in the model as covariate. Based on the result table it probably was and no model selection was done, but could you clarify that. Would you get a different outcome by using structural size estimate rather than body mass for scaling?

-Sorry if the text could be confusing, but it is now clear from the text that all variables stayed in the model including body mass, see lines 162-163. The reason that we included body mass was two-fold. First that we wanted to include body mass because studies have shown that it was a variable that could determine prey size in accordance with i. e. beak size. Second, as the referee mentions brain size can depend on body size, by including body mass, we had reduced the possibility that a relationship between brain size and prey size was a trivial relationship. The reason for not including both body mass and also a structural variable as for example femur length, was that there is a significant relationship between body mass and femur length (t = 4.17; P < 0.0001). For these reasons, we selected body mass for the model.

-As suggested by the referee we try to include femur length as a structural parameter in the model, and found that there was some minor differences in the results depending on the prey type and size was analyzed. However, it did not change the original statistical relationships found. Thus, we did not include femur length in the model. 

Would there be a reason to suspect that YEAR needs to be included in the model (sampling areas, food availability etc.)? If not, would be nice to see explanation included in the statistical analysis description.

-As suggested by the referee we have included YEAR in the model in the revised manuscript, see lines 156-159. The new analysis changed the results in two ways, (a) the importance of head volume in relation to prey size is now more clear, and (b) the relationship between shore crab and gizzard mass as shown in the original manuscript, was no longer statistical significant. Thus, the analysis of the relationship between the size of shore crab and area of webbed feet is no longer relevant. Thus, the paragraph is left out in the revised manuscript.

Would using median for prey size rather than mean influence the results? Median value would perhaps be more appropriate in relation to „optimal“ selection as it would indicate that individual selected such item more frequently.

-This is a good idea, and we have prepared the data. However, preliminary analyses showed highly significant correlation between mean prey size and median prey size. As an example the correlation between mean size and median size of one of the most abundant prey species, the blue mussel, showed a highly significant correlation between the two measurements (r2 = 0.98). For this reason, we assessed that including median size instead of mean size would not change or improve the results. 

As authors already have presence-absence data it could be interesting to see how prey diversity is related to brain size. Are bigger brained individuals going the diversification or the specialization route?

-This is also a good suggestion, and we estimated the diversity using the Shannon Wiener index. However, there was no significant results (or even close to significance) when analyzing the diversity index in relation to the model (body mass, gizzard mass, beak volume, head volume and sex as explanatory variables). Due to this, we have not included these new results in the revised paper.

4. Discussion

I failed to find any comments on possible sex-related differences in prey size selection. Also, what about beak volume, especially given the attention to this trait in the introduction. Please, place your results in relation to your hypothesis and introduction a bit more clearly than it is now.

-Sorry for this forgetfulness. We have now included a paragraph on beak volume and sex-related differences. See lines 196-202.

Minor comments:

Line 31: example of Calidris in the abstract is confusing. This specificity perhaps here is not needed.

-We have deleted the example, and changed the first lines of the abstract, see lines 30-32.

Lines 40-43 This is confusing sentence as it is not clear what were dependent variables in your study. As it is now written it seems that you looked at brain size vs other morphological traits.

-The referee is right, the sentence was confusing. It is now corrected, see lines 38-41.

Line 46: The last sentence of the abstract feels disconnected from previous description of results.

-We have changed the sentence, see lines 46-48.

Line 80 sentence is repetitive

-The sentence is deleted.

Line 83 Something is missing here. Perhaps a connector part on how cognitive abilities are increasingly recognized to be associated with brain size?

-We have added some additional sentences, see lines 76-79.

Line 95-98 a confusing sentence

-It is a redundant sentence, and it have been deleted.

Line 108 „..on the other“

-The sentence is corrected, see line 101.

Line 115 what is the trait „potential“? Should it be „potential for successful reproduction..“?

-Correct, it is changed, see line 105-108.

Line 119 strange confusing sentence that looks to be clipped from Methods section. Few lines down the same information is mentioned.

-The sentence has been deleted.

Line 126 Please mention also if the two sexes (plus juveniles) were roughly equally represented for different years.

-The sex and age classes are now specified, see lines 122-124.

Line 132 what do you mean for age here (juveniles vs sexually mature)? Or was the chronological age recorded?

-This is now specified, see line 129.

Line 136 Please, double-check if the provided reference (10:Zhang et al.) is correct for the formula. Zhang et al. did not measure head volume in their study. Also, check the formatting for the head volume formula.

-The reference has been changed, see line 134.

Line 139 More appropriate term for „head“ here would be „skull“. I agree with authors that it could make sense to measure dry brain weight. However, also in previous studies Jaatinen et al. 2019 wet (?) brain mass was measured with the same results. Perhaps mentioning why dry brain weight was more appropriate option (if that was so), how was it standardized and that there were infact no qualitative differences may be good (in case it is so).

-‘Head’ has been changed to ‘skull’, see line 139.

-We have consistently used head size and brain size in our analyses because they are strongly positively correlated accounting for more than 75% of the variance (e.g. 16, 26, papers on eider and brain). 

Line 168 What statistical programme was used for data analyses?

-We used SAS. 2012. JMP. Version 10.0.2. SAS Institute Inc, Cary, NC. See line 164 and 368.

Line 202 check typo „ether“

-The paragraph is now deleted.

Lines 205-207 These are repetition from result section. Could be better to introduce a general pattern or main finding.

-The referee is right. The sentence is now changed, see lines 187-189.

Lines 216-222 This information actually would have been nice already in the introduction and not here.

-The paragraphs has been moved to the introduction, see lines 109-115.

Line 236 Apologies, but again a confusing sentence. Do you have in mind that you found extreme sizes or that eiders seemed to prefer larger than expected prey?

-The sentence has been corrected, see lines 225-226.

Line 238 what species was this done on? Also word „such as “ missing

-The species is now added, see lines 226-229.

Line 254 but you do not find that. If that is because lab vs natural condition then connection to the previous sentence should be clearer.

-We have now clearly mentioned if we are writing about feeding under natural or experimental conditions, see lines 231 and 243-245.

Line 255-258 very unclear sentence

-The sentence was confusing, and it has been deleted.

Line 259-267 I would try placing this para just before conclusions.

-We have followed the advice and moved parts of this paragraph before the conclusion, see lines 250-254.

Line 275 and 285 Would suggest to avoid „obviously“ and tone down.

-The paragraph has been deleted. 

Reference list not formatted properly.

-We have improved the reference list.

Reviewer #2: The authors should change the title of the paper. For instance, their findings are not only positive relationship between brain volume and prey size but also positive relationship between the size of shore crabs and the area of web of feet.

-The results of relationships between shore crabs and webbed feet have been deleted in the revised version. Thus, the title is not changed.

The authors found positive relationship between head volume and brain mass (F = 39.21, df = 1,13, p < 0.0001, r2 = 0.73). But, can the head volume really be a proxy of the brain size? I agree that the head volume could be one of the indicators of the degree of brain development, though. Better idea would be that the authors use the volume of brain itself (or the size of brain domain specific for visual cognition) as the indicator of cognitive abilities.

-The referee is right that data on the size of the brains for all 198 eiders would have been better for the analysis and probably giving a more precise result than using the size of the heads. However, we emphasize that the two variables are strongly positively correlated accounting for 75% of the variance (Jaatinen et al. 2019). Given that head size and brain size are tightly, positively correlated, it is unlikely that the conclusions would have changed if we had used one rather than the other. 

Introduction Line3-7: Accipritridae, Podicipedidae, and Fringillidae, for example, are all "Family" name of birds. So, the authors should write them in regular style, not in italic style.

-The lines have been deleted.

---

## [Decision Letter · Decision Letter 1]

18 Jan 2021

PONE-D-20-17519R1

Brain size explains prey size selection better than beak, gizzard and body size in a benthivorous duck species

PLOS ONE

Dear Dr. Laursen,

Thank you for submitting your manuscript to PLOS ONE. After careful consideration, we feel that it has merit but does not fully meet PLOS ONE’s publication criteria as it currently stands. Therefore, we invite you to submit a revised version of the manuscript that addresses the points raised during the review process.

We look forward to receiving your revised manuscript.

Kind regards,

Vitor Hugo Rodrigues Paiva, Ph.D.

Academic Editor

PLOS ONE

Reviewers' comments:

Reviewer's Responses to Questions

**Comments to the Author**

1. If the authors have adequately addressed your comments raised in a previous round of review and you feel that this manuscript is now acceptable for publication, you may indicate that here to bypass the “Comments to the Author” section, enter your conflict of interest statement in the “Confidential to Editor” section, and submit your "Accept" recommendation.

Reviewer #1: (No Response)

Reviewer #3: (No Response)

2. Is the manuscript technically sound, and do the data support the conclusions?

Reviewer #1: Yes

Reviewer #3: No

3. Has the statistical analysis been performed appropriately and rigorously? 

Reviewer #1: Yes

Reviewer #3: Yes

4. Have the authors made all data underlying the findings in their manuscript fully available?

Reviewer #1: No

Reviewer #3: Yes

5. Is the manuscript presented in an intelligible fashion and written in standard English?

Reviewer #1: Yes

Reviewer #3: No

6. Review Comments to the Author

Reviewer #1: Response to authors:

I thank authors for answering my questions and making changes to their analyses or/and text. The manuscript seems to be much improved. Yet, I would still have a couple of comments which authors may take into their consideration. Also, I would suggest authors to carefully go through their manuscript to improve style and fluency of the text.

##############################################################

Minor comments:

L 42 „positively“

L60 „differences“ not needed?

L63 “by“ to „with“

LL91-116. The whole paragraphs may still benefit from careful reading of the text. Part of the problem may be change in the tense starting at L96. If you do not want to change the place where study species is introduced, it could be more clearer if you wrote „....Body mass and sex should be included in such analyses as for some bird species these traits are known to have an effect on prey size selection (5,23).” Or similiar. Similar suggestion would be for the next sentence.

L 96 it could be debated if sex and body mass should be called anatomical structures.

L144 should be changed to „...we measured as the total (longest) length of the shell to the nearest XXmm.“ or similar

L145-149 Sentence is difficult to understand. It could be broken into separate ones according to group.

L149 Unclear. Perhaps „Size of broken items was approximated to the nearest 5mm using a size-appropriate reference from collection of intact prey samples“. Would that be what you mean to say?

L150 „For each eider“ or „...individual..“, cause how many gizzards does one individual have?

LL159-162 Sentence should be checked for clarity.

L164 Include, version for JMP if appropriate

L200 It is unclear then which sex took larger prey. In results it is stated that females take smaller prey, while in discussion it becomes no longer that clear.

L206-208 Sentence could be nicely split into two.

L219 I would exclude word „Obviously“ or tone down this sentence. For human senses bivalves may be difficult to find, but we still know surprisingly little about bird senses including for instance olfaction.

L241 If smaller fraction is lost, then your whole mean is shifted towards larger numbers and thus you are „overestimating“ rather than „underestimating“ mean prey size?

LL 257-265 I imagine this part (Conclusions) should contain more general statements than the summary of results.

L413 (Fig 1 legend). Use of „Number (%) ...“ is confusing. Also, in the figure it self, perhaps spelling what is on y-axis would be more clear. Perhaps, "Proportion (%)..."

L424 Table 1.What are the numbers given in Table 1 and how are they related to those given in first lines of the results section. Are numbers provided in this table describe only intact items? If so, then it should be mentioned.

Reviewer #3: Major comments

My main concern is that in its current state, this study shows a correlation of head size with prey item size, which may be just an artifact of the mechanical limit of ingesting big preys (i.e. only big birds with big mouths can ingest the big prey). This idea is supported by the fact that only the larger items show a correlation with head size. Small food items will not show a correlation because they are far from the mechanical limit, contrary to the expected if the eiders select for relative large preys. If the authors want to test if eating big prey items is a cognitively demanding task, a correlation of relative brain size (brain corrected by a measure of overall body size) with relative prey size would be a more suitable approach. If the hypothesis is true, individuals with relatively larger brains will ingest relatively larger items. Moreover, head size probably correlates with gape width too (it could have even a stronger relationship than head-brain size), so it is hard to distinguish between both effects. Therefore, the authors need to tone down their interpretations, this study is completely correlational, and statements such as “Brain size explains prey size selection” and “These novel results indicate that cognitive processes connected to brain size are involved in prey size selection by eiders” (L47-48), should be avoided.

Also, the authors use the term “size” quite often throughout the manuscript, but sometimes it actually has different meanings (e.g. volume, length, mass), which blurs the interpretation. For example, this paper used volume as a measure of beak size, which is different from the measure used in reference 9 (i.e. beak length: L62-63). Using the actual variables, when available, will facilitate the comprehension and transferability of the conclusions.

Minor comments

Abstract

L30 This opening statement is not quite true. Even though this was a common hypothesis in the literature, it does not hold in most studies. The authors do a good job in the introduction showing how this is not always true, therefore this statement should be toned down.

L32 This paper did not use the total brain size. Then, the authors present the actual variable (L37), So I suggest deleting or editing this statement to reflect what is actually done.

L41 Hinia reticulata** (L140 too)

L45 “Strongly” is a bit subjective, your results show a significant effect, but it is not necessarily strong.

Introduction

L59-60 I think this idea describes reference 4 instead of reference 6, and maybe it needs to be rephrased… Diet explains a small fraction of the shape’s variation, instead of a small fraction of the diet explains variation. Right??

L62-63 Is reference 9 based on Calidris canutus or Calidris tenuirostris??? Please revise the reference

L69-73 Do eiders (or other birds) ever eat items as large as the gizzard?? Or this feedback mechanism would affect the number of preys (total volume) rather than the size of each individual item. For instance, the authors found 1299 items in 198 individuals (average 6.5 prey items). This might be important to discuss later.

L80 Does it refers to absolute and relative brain size, rather than “simply brain size”

L98 Head volume is a pretty rough approximation for brain size, authors need to acknowledge that and be careful with the interpretation. For instance, ~25% of the variation in brain size is not explained by head size (L174).

Methods

L121 By convention latitudinal coordinates go before longitudinal coordinates. i.e. (55 ° 50’ N; 10 ° 20’ E)

L135-136 It is unclear to me why authors fitted an ellipsoid on a non-elliptical object (the beak) to get a measure of size (volume), if this a common practice or for comparative purposes, please cite. Wouldn’t be better to combine the three variables into a single one using a PCA to get a measure of “size”??

L148-149 I do not understand how you are using this number later in your results, and why to divide it by four, please clarify. Is this a common practice? Please cite.

L157-… by use of a multivariate GLM. This may help to clarify that all dependent variables are included in a single model.

Results

L181-182 It should say a “significant positive effect” instead of a “high positive effect”.

L183 What is “(0.0193)”?

Discussion

L192 I agree that, historically, beak morphology has been hypothesized to have a direct relationship with food item characteristics. This is particularly true when the beak is used to manipulate the food, like finches cracking strong seeds. However, several studies have shown that species that ingest the items whole (as is the case of the common eider), the item size is limited or correlated with gape width (e.g. Hulsman 1981, Wheelwright 1985, Saunders et al. 1995, Dehling et al. 2016). This should be included in the discussion.

L194-196 Perhaps rephrase this: “These results support previous findings that, in general, other morphological traits than beak size drives prey size selection [4,6]”

L200 To me, it seems advantageous for both sexes to build up body condition during winter, not just for females. Authors may argue that females are under stronger pressure due to nesting. However, the results showed that females eat smaller prey items (L178-179).

L219 I would suggest avoiding “obviously” on a scientific manuscript and tone down this claiming. Especially, if it is unknown how easily the eiders would localize the bivalves by the siphons.

L225 “larger prey size than expected” Where this expectation come from? This does not seem to be a prediction of this study. Is this based on literature?? Please clarify.

L239-241 If smaller items are digested quickly, your sample would be skewed to the larger items. Thus, you are likely overestimating prey size

L246 “At the breeding grounds, eiders demonstrate complex behavior.”: Complexity is a tricky term and difficult to define. Also, this sentence is a bit vague and not directly linked to your results or the following ideas. I suggest modifying it.

L246-249 This paragraph seems disconnected. How is this paragraph linked to your project?

L250-252 Shouldn't this state something like “seasonal variation in resources and conditions” instead of “the studies”? Perhaps a quick recapitulation of the main evidence would clarify how do you get to this idea.

L263-265 These results showed that individuals with large heads eat larger prey items, and in general, individuals with larger heads do also have larger brains. However, this pattern could be easily explained by a size constrain on prey size-gape width.

In general, the manuscript needs to be carefully read and edited, for grammatical and punctuation errors. For example (but not limited to)

L44 missing a period

L61 missing a comma after “Within species”

L183 missing a closing parenthesis

L191 missing a comma

L203 double tab

7. PLOS authors have the option to publish the peer review history of their article (what does this mean?). If published, this will include your full peer review and any attached files.

Reviewer #1: No

Reviewer #3: No

---

## [Author Response · Author response to Decision Letter 1]

28 Feb 2021

Reviewer #1: Response to authors:

I thank the authors for answering my questions and making changes to their analyses or/and text. The manuscript seems to be much improved. Yet, I would still have a couple of comments which authors may take into their consideration. Also, I would suggest authors to carefully go through their manuscript to improve style and fluency of the text.

_Thank you for the comments, which we have taken into account.

##############################################################

Minor comments:

L 42 „positively“

_Corrected.

L60 „differences“ not needed?

_Deleted.

L63 “by“ to „with“

_Corrected.

LL91-116. The whole paragraphs may still benefit from careful reading of the text. Part of the problem may be change in the tense starting at L96. If you do not want to change the place where study species is introduced, it could be more clearer if you wrote „....Body mass and sex should be included in such analyses as for some bird species these traits are known to have an effect on prey size selection (5,23).” Or similiar. Similar suggestion would be for the next sentence.

_We have included the sentence suggested by the referee about body mass and sex, and changed the sentence accordingly. See lines 97-99.

L 96 it could be debated if sex and body mass should be called anatomical structures.

_The referee has a point here. The sentence is changed. See lines 96-97.

L144 should be changed to „...we measured as the total (longest) length of the shell to the nearest XXmm.“ or similar

_We have changed the description of assessing the size of the food items. See lines 154-157.

L145-149 Sentence is difficult to understand. It could be broken into separate ones according to group.

_The paragraph is changed. See lines 151-158.

L149 Unclear. Perhaps „Size of broken items was approximated to the nearest 5mm using a size-appropriate reference from collection of intact prey samples“. Would that be what you mean to say?

_Thank you for the suggestion. See l. 156.

L150 „For each eider“ or „...individual..“, cause how many gizzards does one individual have?

_Corrected.

LL159-162 Sentence should be checked for clarity.

_Corrected. The sentence was unclear. It is corrected. See lines 168-169.

L164 Include, version for JMP if appropriate

_We have added the version. See l. 173.

L200 It is unclear then which sex took larger prey. In results it is stated that females take smaller prey, while in discussion it becomes no longer that clear.

_Sex differences in prey size are interesting, but not an important point here. A reason to include sex in the statistical analysis is to reduce the variance in the data.

L206-208 Sentence could be nicely split into two.

_Thank you for the suggestion. See lines 227-230.

L219 I would exclude word „Obviously“ or tone down this sentence. For human senses bivalves may be difficult to find, but we still know surprisingly little about bird senses including for instance olfaction.

_We have done so.

L241 If smaller fraction is lost, then your whole mean is shifted towards larger numbers and thus you are „overestimating“ rather than „underestimating“ mean prey size?

_Sorry, of course the referee is right. It is corrected. See l. 264.

LL 257-265 I imagine this part (Conclusions) should contain more general statements than the summary of results.

_We have changed the wording. See lines 276-278.

L413 (Fig 1 legend). Use of „Number (%) ...“ is confusing. Also, in the figure it self, perhaps spelling what is on y-axis would be more clear. Perhaps, "Proportion (%)..."

_It is corrected. See l. 427.

L424 Table 1.What are the numbers given in Table 1 and how are they related to those given in first lines of the results section. Are numbers provided in this table describe only intact items? If so, then it should be mentioned.

_The referee has a point. It is now add in Tabel 1 that N is the number of eiders with a given prey type. Opposite to the first line in the Result Section, which is based on the total number of prey. See lines 442 and 177.

Reviewer #3: Major comments

My main concern is that in its current state, this study shows a correlation of head size with prey item size, which may be just an artifact of the mechanical limit of ingesting big preys (i.e. only big birds with big mouths can ingest the big prey). This idea is supported by the fact that only the larger items show a correlation with head size. Small food items will not show a correlation because they are far from the mechanical limit, contrary to the expected if the eiders select for relative large preys. If the authors want to test if eating big prey items is a cognitively demanding task, a correlation of relative brain size (brain corrected by a measure of overall body size) with relative prey size would be a more suitable approach. If the hypothesis is true, individuals with relatively larger brains will ingest relatively larger items. Moreover, head size probably correlates with gape width too (it could have even a stronger relationship than head-brain size), so it is hard to distinguish between both effects. Therefore, the authors need to tone down their interpretations, this study is completely correlational, and statements such as “Brain size explains prey size selection” and “These novel results indicate that cognitive processes connected to brain size are involved in prey size selection by eiders” (L47-48), should be avoided.

_Two important aspect were raised here by the referee. 

_First, about the correlations. It is correct that if the overall size of the eiders was not included in the analyses, we would just get a trivial correlation showing that eiders with big heads took large prey. However, we had included body size (as body mass) in all analyses, to eliminate this effect. 

_Second, the gape width is a good point raised by the referee. To be honest, we have not been thinking about the aspect of gape width. However, it sounds reasonable that the gape could set limits for the prey size taken, even that the hinge between the upper and lower mandible is flexible. We had an opportunity to examine the gape of 21 eiders sampled in February 2021. The mean (se) of the gape width was 44.4 (0.6) mm, the gape height 39.3 (0.9) mm. From our reference sample of intact benthos species, we measured length and width relations of blue mussels, razor clams, cockles, conch and shore crab. From our data file of diet, the largest blue mussel taken by the eider was 66 mm, for cockle 45 mm, for razor clam 148 mm, for conch 68 mm and for shore crab 55 mm. The corresponding width for the longest blue mussel is 28 mm, for cockle 42 mm, for razor clam 22 mm, for conch 40 mm and for shore crab 38 mm. The measurements for all food items are smaller than the mean width of the gape (44.4 mm). From this, we considered it unlikely that the dimensions of the gape should restrict intake of food items in eiders. We have included a short summary in the revised paper, as an alternative hypothesis. See lines 118-121, 158-159 and 194-206. 

Also, the authors use the term “size” quite often throughout the manuscript, but sometimes it actually has different meanings (e.g. volume, length, mass), which blurs the interpretation. For example, this paper used volume as a measure of beak size, which is different from the measure used in reference 9 (i.e. beak length: L62-63). Using the actual variables, when available, will facilitate the comprehension and transferability of the conclusions.

_Sorry for using ‘size’ as a general expression for a dimension. It was done simply to make the reading more straightforward and easy. We have changed the wording accordingly. However, we will still use beak volume as a measurement for beak dimension, since in our opinion it makes the statistical analyses simpler (one variable instead of three variables). 

Minor comments

Abstract

L30 This opening statement is not quite true. Even though this was a common hypothesis in the literature, it does not hold in most studies. The authors do a good job in the introduction showing how this is not always true, therefore this statement should be toned down.

_We have changed the sentence. See lines 30 and 55. 

L32 This paper did not use the total brain size. Then, the authors present the actual variable (L37), So I suggest deleting or editing this statement to reflect what is actually done.

_We have skipped ‘total’ and changed ‘brain size’ to ‘brain mass’, which was actually measured. Brain mass is used throughout in the revised paper. See lines 32, 37, 38 and 47.

L41 Hinia reticulata** (L140 too).

_Sorry, this is now corrected, see lines 41 and 146. 

L45 “Strongly” is a bit subjective, your results show a significant effect, but it is not necessarily strong.

_It is deleted. See l. 45.

Introduction

L59-60 I think this idea describes reference 4 instead of reference 6, and maybe it needs to be rephrased… Diet explains a small fraction of the shape’s variation, instead of a small fraction of the diet explains variation. Right??

_The referee is right. The sentence is changed. See l. 60-62.

L62-63 Is reference 9 based on Calidris canutus or Calidris tenuirostris??? Please revise the reference

It is correct. The species name has been changed. See l. 63.

L69-73 Do eiders (or other birds) ever eat items as large as the gizzard?? Or this feedback mechanism would affect the number of preys (total volume) rather than the size of each individual item. For instance, the authors found 1299 items in 198 individuals (average 6.5 prey items). This might be important to discuss later.

_There are some good point raised here by the referee. 

_First, the prey size in relation to gizzard size. At some occasions, eiders have taken razor clams larger than the gizzard. One end of the clam was in the gizzard and the other end in the esophagus. The reason to take such a large prey could be that it represents a large amount of flesh and that it is soft shelled. We have added a sentence about these observations. See lines 247-249. 

_Second, the question if the amount of food in the gizzard has a sort of feedback mechanism on selection on each item. This is an interesting question, which we have not considered, to be frank. However, the question cannot be answered directly, because we don’t know in which order the prey items in the gizzard have been taken. However, we can get an impression from our data. If there is a sort of feedback, we should expect few and large preys in the gizzards in contrast to many and small preys. The data reviles the opposite result, that there was a positive relationship between the number of preys and the mean prey size in the gizzards (r2 = 0.26) and for the maximum prey size (r2 = 0.22). Which does not support the assumption. Due to these points being marginal, we will not pursue this item further.

L80 Does it refers to absolute and relative brain size, rather than “simply brain size”

_This is now corrected for brain mass. See l. 81.

L98 Head volume is a pretty rough approximation for brain size, authors need to acknowledge that and be careful with the interpretation. For instance, ~25% of the variation in brain size is not explained by head size (L174).

_The referee is right. On the other hand, in most ecological studies only about 8% of the variation was explained (Oecologia 132, 492-500). 

Methods

L121 By convention latitudinal coordinates go before longitudinal coordinates. i.e. (55 ° 50’ N; 10 ° 20’ E)

_Sorry, of course. It is corrected. See l. 125.

L135-136 It is unclear to me why authors fitted an ellipsoid on a non-elliptical object (the beak) to get a measure of size (volume), if this a common practice or for comparative purposes, please cite. Wouldn’t be better to combine the three variables into a single one using a PCA to get a measure of “size”??

_We used estimates of volumes when calculating the size of the head (other variables and prey (razor clam for example)). We did so because previous studies of brain size have been based on ellipsoids (2 radii of similar size and one that is larger) rather than circular volumes (3 radii of similar size). Proc. R. Soc. Long. B; (2002) 269:961-967. PLOS One (2020) 15 (9): e0236155. 

_The use of Principal Component Analyses based on the correlation matrix from the log-transformed data, we still do not find any evidence consistent with the expectations suggested by the reviewer, and only one of the relationships reached statistical significance. 

L148-149 I do not understand how you are using this number later in your results, and why to divide it by four, please clarify. Is this a common practice? Please cite.

_It is simply to get the number of crabs. They have four claws, and when we count four claws of the same size the eider have taken (at least) one crab. We have corrected the text. See line 154. 

L157-… by use of a multivariate GLM. This may help to clarify that all dependent variables are included in a single model.

_Thank you, it is now added to the text. See l. 166.

Results

L181-182 It should say a “significant positive effect” instead of a “high positive effect”.

_Yes of course. It is corrected. See lines. 182-183.

L183 What is “(0.0193)”?

_It is the ‘se’ value. It is now corrected. See l. 192.

Discussion

L192 I agree that, historically, beak morphology has been hypothesized to have a direct relationship with food item characteristics. This is particularly true when the beak is used to manipulate the food, like finches cracking strong seeds. However, several studies have shown that species that ingest the items whole (as is the case of the common eider), the item size is limited or correlated with gape width (e.g. Hulsman 1981, Wheelwright 1985, Saunders et al. 1995, Dehling et al. 2016). This should be included in the discussion.

_We have included gab size in the revised manuscript. See lines 118-121, 158-159 and 194-206.

L194-196 Perhaps rephrase this: “These results support previous findings that, in general, other morphological traits than beak size drives prey size selection [4,6]”

_Thank you for this suggestion. The sentence has been used. See lines 216-217.

L200 To me, it seems advantageous for both sexes to build up body condition during winter, not just for females. Authors may argue that females are under stronger pressure due to nesting. However, the results showed that females eat smaller prey items (L178-179).

_Thank you for this suggestion. It is included. See lines 222-224.

L219 I would suggest avoiding “obviously” on a scientific manuscript and tone down this claiming. Especially, if it is unknown how easily the eiders would localize the bivalves by the siphons.

_The word has been deleted, and the text toned down. See lines 240-242.

L225 “larger prey size than expected” Where this expectation come from? This does not seem to be a prediction of this study. Is this based on literature?? Please clarify.

_The sentence has been changed. See l. 246.

L239-241 If smaller items are digested quickly, your sample would be skewed to the larger items. Thus, you are likely overestimating prey size.

_Yes of course. It has been changed. See l. 264.

L246 “At the breeding grounds, eiders demonstrate complex behavior.”: Complexity is a tricky term and difficult to define. Also, this sentence is a bit vague and not directly linked to your results or the following ideas. I suggest modifying it.

_The sentence have been deleted.

 L246-249 This paragraph seems disconnected. How is this paragraph linked to your project?

_The paragraphs were written trying to give a full picture of eider ecology during winter and breeding. However, the referee is right, it is not connected to the study. Thus, it is deleted.

L250-252 Shouldn't this state something like “seasonal variation in resources and conditions” instead of “the studies”? Perhaps a quick recapitulation of the main evidence would clarify how do you get to this idea.

_Thank you for the suggestion. The sentence has been changed. See lines 269-271.

L263-265 These results showed that individuals with large heads eat larger prey items, and in general, individuals with larger heads do also have larger brains. However, this pattern could be easily explained by a size constrain on prey size-gape width.

_Gape size has been included in the study. See lines 118-121, 158-159 and 194-206. 

_The Conclusion Section has been shortened. See lines 276-279.

In general, the manuscript needs to be carefully read and edited, for grammatical and punctuation errors. For example (but not limited to)

L44 missing a period

L61 missing a comma after “Within species”

L183 missing a closing parenthesis

L191 missing a comma

L203 double tab

_Thank you for the advice and for the corrections.

---

## [Editor Report · Decision Letter 2]

3 Mar 2021

Brain mass explains prey size selection better than beak, gizzard and body size in a benthivorous duck species

PONE-D-20-17519R2

Dear Dr. Laursen,

We’re pleased to inform you that your manuscript has been judged scientifically suitable for publication and will be formally accepted for publication once it meets all outstanding technical requirements.

Kind regards,

Vitor Hugo Rodrigues Paiva, Ph.D.

Academic Editor

PLOS ONE
---

## [Editor Report · Acceptance letter]

18 Mar 2021

PONE-D-20-17519R2 

Brain mass explains prey size selection better than beak, gizzard and body size in a benthivorous duck species 

Dear Dr. Laursen:

I'm pleased to inform you that your manuscript has been deemed suitable for publication in PLOS ONE. Congratulations! Your manuscript is now with our production department. 

Kind regards, 

on behalf of

Dr. Vitor Hugo Rodrigues Paiva 

Academic Editor

PLOS ONE